# Effect of *Holoptelea integrifolia* (Roxb.) Planch. *n*-Hexane Extract and Its Bioactive Compounds on Wound Healing and Anti-Inflammatory Activity

**DOI:** 10.3390/molecules27238540

**Published:** 2022-12-04

**Authors:** Kanokwan Somwong, Pattawika Lertpatipanpong, Wutigri Nimlamool, Aussara Panya, Yingmanee Tragoolpua, Rujipas Yongsawas, Wandee Gritsanapan, Hataichanok Pandith, Seung Joon Baek

**Affiliations:** 1Department of Biology, Faculty of Science, Chiang Mai University, Chiang Mai 50200, Thailand; 2Laboratory of Signal Transduction, College of Veterinary Medicine and Research Institute for Veterinary Science, Seoul National University, Seoul 08826, Republic of Korea; 3Department of Pharmacology, Faculty of Medicine, Chiang Mai University, Chiang Mai 50200, Thailand; 4Research Center in Bioresources for Agriculture, Industry and Medicine, Chiang Mai 50200, Thailand; 5Department of Pharmacognosy, Faculty of Pharmacy, Mahidol University, Bangkok 10400, Thailand

**Keywords:** *Holoptelea integrifolia*, MMP-9, anti-inflammation, wound healing, friedelin

## Abstract

The stem bark of *Holoptelea integrifolia* (Roxb.) Planch. has been applied for the treatment of human cutaneous diseases as well as canine demodicosis in several countries. However, no detailed mechanistic studies have been reported to support their use. In this study, thin-layer chromatography and gas chromatography were used to screen phytochemicals from the fresh stem bark extract of *H. integrifolia*. We found the two major bioactive compounds, friedelin and lupeol, and their activity on wound healing was further investigated in keratinocytes. Both bioactive compounds significantly reduced wound area and increased keratinocyte migration by increasing matrix metalloproteinases-9 production. Subsequently, we found that the mRNA gene expressions of cadherin 1 and desmoglobin 1 significantly decreased, whereas the gene expression involved in keratinocyte proliferation and homeostasis (keratin-17) increased in compound-treated human immortalized keratinocytes cells. The expression of inflammatory genes (cyclooxygenase-2 and inducible nitric oxide synthase) and pro-inflammatory cytokine genes (tumor necrosis factor-alpha and interleukin-6) was reduced by treatment with *n*-hexane extract of *H. integrifolia* and its bioactive compounds. Our results revealed that *H. integrifolia* extract and its bioactive compounds, friedelin and lupeol, exhibit wound-healing activity with anti-inflammatory properties, mediated by regulating the gene expression involved in skin re-epithelialization.

## 1. Introduction

*Holoptelea integrifolia* (Roxb.) Planch. is a medicinal plant that is traditionally used for the treatment and prevention of several diseases. The bark and leaves of this plant are the most investigated parts and have anti-inflammatory effects, and their extracts are externally applied to treat several skin diseases, such as leprosy, leukoderma, and chronic wounds [1,2]. The stem bark extract of this plant contains various phytochemicals, such as terpenoids, friedelin, β-amyrin, betulin, betulinic acid, and lupeol [3].

Cutaneous wound healing is a complex biological process that can be classified into three major phases: inflammation, proliferation, and maturation phase [4,5]. Re-epithelialization during the proliferative phase is important for successful wound closure. Matrix metalloproteinases (MMPs) play a crucial role in augmenting keratinocyte migration by degrading extracellular matrix proteins secreted by keratinocytes in the neo-epidermis. Among these, MMP-1 and MMP-9 have been reported to function as mediating enzymes for keratinocyte migration, and MMP-9 in particular promotes terminal cell differentiation [6]. Moreover, inflammation can occur immediately after skin injury, leading to the upregulation of several pro-inflammatory cytokines, including cyclooxygenase-2 (COX-2) and inducible nitric oxide synthase (iNOS) that play a critical role in inflammation. These enzymes are triggered to produce pro-inflammatory mediators, thereby enhancing the expression of pro-inflammatory cytokines, including tumor necrosis factor-alpha (TNF-α) and interleukin-6 (IL-6).

Various herbal drugs and formulations have been widely used in the treatment of wounds for years as several natural products, such as alkaloids, tannins, flavonoids, and terpenes promote the wound healing process. However, traditional herbal medicinal products for wound healing require elucidation of their molecular mechanisms. This will provide beneficial clues for accelerating wound healing and encourage fighting against skin infection.

Several studies have reported the wound-healing activity of *H. integrifolia*. For example, the wound-healing activity of *H. integrifolia* extracts was examined in excision wound model in Wister rats, the external application of these extracts on the wound sites show more than 90% wound healing was recorded in treated groups by 14 days of post-surgery, whereas only 62.99% was observed in the control group albino rats. In addition, in incision model, the higher collagen re-deposition than the control group was presented in treated group. Not only wound healing property, but also prevented microbes from invading the wound, protecting it against microbial infections [7,8].

In this study, we extracted the fresh stem bark of *H. integrifolia* using various solvents and investigated the bioactive components in these extracts for keratinocyte wound healing and anti-inflammatory activity. This is the first study suggesting that *H. integrifolia* stem bark extracts and their active compounds modulate the expression of re-epithelialization-related genes and MMP-9, thereby enhancing wound-healing activity.

## 2. Results

### 2.1. Effects of H. Integrifolia n-Hexane Extract on Cell Viability in Human and Canine Keratinocytes

According to the TLC result (Appendix A), among those various solvent, *n*-hexane extract provides highest content of phytochemical components after extraction. Therefore, *n*-hexane extract was selected to use in this study. Human and canine keratinocytes were treated with various concentrations of *H. integrifolia n*-hexane extract (1–20 μg/mL), and cell viability was evaluated using the CellTiter 96^®^ assay kit. After 24 h of treatment, the viability of human keratinocytes significantly decreased to 57.9% and 31.4% at 10 μg/mL and 20 μg/mL, respectively (Figure 1A), while it decreased to 70.8% and 3.87%, respectively, after 48 h of treatment. Similarly, the cell viability of canine keratinocytes significantly decreased to 27.0% and 18.1% at 10 and 20 μg/mL *n*-hexane extracts, respectively, after 24 h of treatment (Figure 1B), whereas it was completely abolished in both 10 and 20 μg/mL after 48 h of treatment. Subsequently, 1 μg/mL *n*-hexane extract was selected, and the scratch wound healing assay was conducted using human keratinocytes. Results showed that the wound area was marginally affected by the *n*-hexane extract; however, no significant difference in the percentage of wound area between the DMSO control and the *n*-hexane extract treatment was observed (Figure 1C).

### 2.2. Identification of Friedelin and Lupeol in n-Hexane Extract

*H. integrifolia* extract contains several phytochemicals, including friedelin and lupeol [3,9,10]. The *n*-hexane extract was analyzed using TLC, and the developed TLC indicated that two triterpenoids, friedelin (Rf = 0.9625, 0.9125) and lupeol (Rf = 0.8625), were the major components in the extract (Figure 2A). Subsequently, the chromatogram of GC-FID analysis showed the following compounds: lupeol (RT 97) and friedelin (RT 104) (Figure 2B), which had the standard specific peaks of each compound. Thus, friedelin and lupeol were the major phytochemical components found in the *n*-hexane extract of the fresh stem bark of *H. integrifolia*.

### 2.3. Effect of Friedelin and Lupeol on Cell Viability and Wound-Healing Activity in Keratinocytes

Human and canine keratinocytes were treated with various concentrations of friedelin (0.01 to 4 μM) and lupeol (0.2 to 20 μM). The cell viability results showed that the percentage of cell survival in 4 μM friedelin-treated HaCaT cells decreased to 57.7% and 65.3% at 24 and 48 h, respectively (Figure 3A, left); 20 μM lupeol-treated HaCaT cells viability decreased to 76.0% and 71.15% at 24 and 48 h, respectively. For the canine keratinocytes, 0.4 μM friedelin reduced canine keratinocyte viability to 49.3% and 59.2% at 24 and 48 h, respectively (Figure 3B, left). Lupeol at concentration 20 μM decreased canine keratinocytes cell viability to 63.1% and 54.7%, respectively, while lupeol at concentration 2 μM decreased it to 82.1% (Figure 3B, right). A scratch assay was conducted using friedelin and lupeol at non-toxic doses. Results revealed that friedelin (0.04 μM) and lupeol (0.2 μM) reduced the percentage of wound area by 78.84% (*p* < 0.01) and 11.17% (*p* < 0.001) at 48 h, respectively, compared to the DMSO treatment (Figure 3C). Thus, friedelin and lupeol treatment may facilitate wound closure after 48 h of treatment.

### 2.4. MMPs Expression by Friedelin

The MMPs are well-known proteins that play important roles in wound-healing activity [11,12]. To investigate the effect of friedelin on proteins belonging to the MMP family involved in keratinocyte re-epithelialization, a human MMP array was performed. As shown in Figure 4A, friedelin treatment increased most MMPs and TIMPs, with the highest increase in the density of MMP-9 protein (2.36-fold increase) compared with the control (DMSO) treatment. The quantitative density of the protein was plotted as fold change compared to that of the control treatment (Figure 4B). To confirm the antibody array data, friedelin (0.04 μM) and lupeol (0.2 μM) were treated into HaCaT cells and the conditioned medium was collected to determine the activity of MMP-9 using gelatin zymography assay. TGF−β1 (0.5 μM) treatment was used as a positive control and the DMSO treatment was used as a negative control. As shown in Figure 4C, the activity of MMP-9 was significantly increased using lupeol (0.2 μM) treatment to 3.54-fold (*p* < 0.01) after 24 h and 4.66-fold (*p* < 0.0001) after 48 h, respectively, in a dose-dependent manner, when compared with the negative control. However, following friedelin treatment, a significant increase in MMP-9 activity (3.1-fold) was observed only after 48 h of treatment.

### 2.5. Effect of Friedelin/Lupeol on Gene Expression in Human Keratinocyte Re-Epithelialization

Desmosomes, adhesions, and structural components, cadherin 1 (CDH-1) and desmoglobin 1 (DSG-1), contribute to keratinocyte migration [13]. In addition, keratin-17 (KRT-17) has been associated with wound-healing activity in keratinocyte proliferation [14,15]. HaCaT cells were treated with 0.04 μM friedelin or 0.2 μM lupeol for 24 and 48 h, respectively, and the RNA expression of CDH-1, DSG-1, and KRT-17 was evaluated to determine the compound-altered gene expression. As shown in Figure 5, friedelin-treated cells significantly decreased CDH-1 gene expression by 0.27-fold (*p* < 0.0001) and 0.56-fold (*p* < 0.001) compared with the DMSO control at 24 and 48 h, respectively, whereas lupeol-treated cells at 48 h showed a significant decrease in CDH-1 expression to 0.29-fold (*p* < 0.0001). DSG-1 gene expression significantly decreased after 48 h of treatment with friedelin, however, not in lupeol-treated cells. The expression of KRT-17 was significantly increased by 2.75-fold (*p* < 0.01) at 48 h after lupeol treatment, compared with the DMSO control.

### 2.6. Effect of H. integrifolia n-Hexane Extract and Friedelin/Lupeol on Anti-Inflammatory Activity in RAW 264.7 Cells

Wound-healing activity is associated with the production of inflammatory cytokines and their clearance by macrophages [16]. To determine whether the extract and its bioactive compounds exhibited anti-inflammatory activity, LPS-induced RAW macrophage cells were used as a positive control. The n-hexane extracts (2.5 and 25 μg/mL) were used to treat the LPS-induced RAW cells. qRT-PCR results indicated that expression of genes involved in two inflammatory enzymes, COX-2 and iNOS, and two pro-inflammatory cytokines, TNF-α and IL-6, were downregulated in extract-treated cells compared to the positive control LPS-treated cells (Figure 6A). Similarly, 1 and 20 μM friedelin/lupeol treatment resulted in reduced COX-2, iNOS, TNF-α, and IL-6 gene expression (Figure 6B). To confirm the qRT-PCR data, the protein expression of the inflammatory enzymes, COX-2 and iNOS, was evaluated using Western blotting. As shown in Figure 6C, friedelin and lupeol treatment decreased LPS-induced COX-2 and iNOS expression in a dose-dependent manner.

## 3. Discussion

The bark of *H. integrifolia* (Roxb.) Planch is used for the treatment of inflammatory diseases and applied dermally on wound lesions by Indian tribes [1]. Triterpenes/triterpenoids and sterols are the major bioactive compounds identified in this plant; however, many other compounds may be biologically active. Although *H. integrifolia* (Roxb.) Planch has potential for wound healing [7] and in vivo anti-inflammatory activity [17,18,19], the molecular mechanisms by which it affects the wound healing process remain unclear. Therefore, in this study, we demonstrated that increased MMP-9 activity, modulated gene expression involved in re-epithelization, and anti-inflammatory activity are key players in the wound-healing activity of *H. integrifolia.*

We identified friedelin and lupeol as the phytochemical components present in the *n*-hexane extract. Friedelin (or friedelan-3-one) has also been identified in the non-polar soluble fraction of *H. integrifolia* stem bark extract [1,9] and is a potent anti-inflammatory compound [20,21]. Lupeol is commonly found in many plants, including *H. integrifolia* stem bark extracts [3,22]. The biological activity of lupeol has anti-oxidizing, anti-cancerous, anti-inflammatory, and wound healing properties [23,24]. Thus, friedelin and lupeol may be promising compounds for wound-healing activity and elucidating their molecular mechanism may contribute to the development of wound healing applications.

A critical process in wound healing is re-epithelialization, which shows evident activity in the proliferative phase and is continuously active until the extracellular matrix remodeling phase. Chronic non-healing wounds can develop because of unsuccessful re-epithelialization. The scratch assay represents cell proliferation and migration [25], and our results suggest that *n*-hexane extract (1 μg/mL) does not significantly reduce wound scratches. However, friedelin and lupeol significantly reduced the in vitro wound scratches (Figure 3). This could be explained by the fact that 1 μg/mL of the extracts used in our study showed insufficient dose to exhibit wound-healing activity, whereas friedelin and lupeol are required at nanomolar concentrations to exhibit wound-healing activity.

During the early progression of wound healing, MMPs production, secreted by keratinocytes, is increased. MMPs initiate the degradation of the extracellular matrix to allow keratinocyte migration. We evaluated the effects of the extract and its active compounds on proteins in the MMP family using a human MMP array. Results revealed that friedelin treatment caused the highest increase in MMP-9 compared to the control treatment (Figure 4A,B). Several studies have indicated that MMP-1 and MMP-9 secreted from migrating keratinocytes promote cell migration and re-epithelialization by degrading cell–cell adhesion proteins [26]. Keratinocytes are the main constituents of the epidermis that re-form the epidermal layer at the wound site and secrete MMP-9 during wound healing progression [5]. Gelatin zymography results demonstrated that MMP-9 activity was increased by friedelin and lupeol treatment (Figure 4C), thereby enhancing re-epithelialization.

In addition, at the middle and later stages of wound repairing, keratinocytes detach from each other to allow cell migration and re-form the epidermal layer at the wound site. The expression of *KRT-17* increases cell growth, resulting in cell hyperproliferation during wound healing [27]. A previous study has suggested that *KRT-17* contributes to the restoration of epidermal permeability barrier in keratinocyte homeostasis [28]. Our results demonstrated altered expression of *KRT-17* with other genes (*CDH1* and *DSG-1*) in friedelin- and lupeol-induced keratinocytes, confirming the role of friedelin/lupeol in re-epithelialization during the healing process.

Inflammation is a critical complication of tissue injury and wound lesions, which can lead to serious conditions. Macrophages play crucial roles in inflammatory stimuli by producing proinflammatory cytokines. Downregulated transcription of proinflammatory genes results in a decrease in the inflammatory response [29]. Therefore, to investigate the effects of *H. integrifolia n*-hexane extract and its bioactive compounds on inflammation, we evaluated the inflammatory enzymes and cytokines. Our results revealed the beneficial effects of *n*-hexane extract and a single compound in LPS-induced RAW cells by downregulating the expression of associated inflammatory genes. Our results are consistent with those of previous studies, indicating that lupeol downregulates the gene expression of TNF-α and IL-6 [30,31]. In addition, 20 μM lupeol reduced the protein expression of COX-2 and iNOS, which induced an inflammatory state to increase prostaglandin production.

Similar to the observations of *n*-hexane extracts, the single bioactive compounds, friedelin and lupeol, illustrated notable benefits for wound healing and anti-inflammatory activity. A detailed study of the pharmacology and action mechanism of *H. integrifolia* may help in understanding the relationship between its pharmacological effects and traditional uses in the future.

In conclusion, our results demonstrated that the fresh stem bark extract of *H. integrifolia* and its single bioactive compounds, friedelin and lupeol, exhibited wound-healing activity via alteration of gene expression related to re-epithelialization, increased MMP-9 activity, and enhanced anti-inflammatory activity.

## 4. Materials and Methods

### 4.1. Reagents and Cell Lines

Lupeol (#L5632) and friedelin (#92187) were purchased from Sigma-Aldrich (St. Louis, MA, USA). Human immortalized keratinocyte (HaCaT) cells were purchased from Cell Line Service (Eppelhein, Germany), and canine immortalized keratinocyte (CPEK) cells were kindly provided by Dr. Cheol-Yong Hwang (College of Veterinary Medicine, Seoul National University). The murine macrophage cell line, RAW 264.7, was purchased from American Type Culture Collection (ATCC, Manassas, VA, USA). All cell lines were maintained in Dulbecco’s modified Eagle medium (DMEM, Gibco, Grand Island, NY, USA) containing 10% fetal bovine serum (FBS), supplemented with 100 U/mL penicillin and 100 mg/mL streptomycin (Gibco), at 37 °C in an incubator with a humidified atmosphere of 5% CO_2_.

### 4.2. Plant Preparation and Extraction

Fresh bark of *H. integrifolia* was collected from Soem Ngam, Lampang, in the northern part of Thailand from March 2019 to August 2019. All specimens were identified by Dr. Narin Printarakul, a taxonomist at the Department of Biology, Faculty of Science, Chiang Mai University. Voucher specimen numbers HI001-003 were maintained at the Chiang Mai University Herbarium. The plant samples were chopped and grounded into small pieces and then separately macerated with various solvents, including *n*-hexane, dichloromethane, ethanol, methanol, and deionized water in a ratio of 1:10 *w*/*v* on a shaker at room temperature overnight. The macerated solution was filtered using Whatman filter paper (Thermo Fisher Scientific, Waltham, MA, USA) before evaporation on a rotary evaporator or water bath for water extraction. The crude extracts were then stored at 4 °C.

### 4.3. Thin-Layer Chromatography (TLC)

TLC was performed with fresh stem bark extracts of *H. integrifolia* and compared to phytochemicals, such as terpenoids, flavonoids, and phenolic compounds; 1 mg/mL of each extract was spotted on a TLC sheet (TLC silica gel 60F245, Merck KGaA, Darmstadt, Germany) with toluene/ethyl acetate/ethanol (8:6:1, *v*/*v*). The TLC sheet was sprayed with anisaldehyde-sulfuric acid reagent (0.5% *p*-anisaldehyde, 10% glacial acetic acid, 85% methanol, and 5% sulfuric acid). After air-drying, the sheet was heated at 100 °C for 5–10 min.

### 4.4. Gas Chromatography Analysis (GC-FID)

The *n*-hexane extract and standard compounds were analyzed using a gas chromatograph (Agilent 7890 B, Agilent Technologies, Santa Clara, CA, USA). The sample (1 μL) was injected into the column and split at a ratio of 1:10 (*v*/*v*) at 250 °C. The injected volume was separated in HP-5 column (length 30 m, diameter 0.320 mm, and film thickness 0.25 µm) at 100 °C, then increased by 4 °C/min until 270 °C and held at 270 °C for 125 min. The flame ionization detector was programmed to operate at 270 °C. The chromatogram retention-time result of the plant extract and a single compound was compared.

### 4.5. Cell Viability Assay

All cell lines were seeded at 5000 cells/well in a 96 well-plate. The cells were then treated with various concentrations of plant extracts and single compounds for 24 h. The cell viability assay was performed using the CellTiter 96^®^ Aqueous One Solution Cell Proliferation Assay kit (Promega, Madison, WI, USA). The optical density was measured using a Multiskan^TM^ FC microplate spectrophotometer (Thermo Fisher Scientific) at a wavelength of 492 nm. Cell viability was then calculated and compared with that of the control group.

### 4.6. Immunoblotting Analysis

The treated cells were washed twice with cold phosphate-buffered saline (PBS), and proteins were extracted using RIPA cell lysis buffer (#R4100, GenDEPOT, TX, USA) with proteinase inhibitor (#BPI001, Biomax, Seoul, Republic of Korea). The protein concentration was measured using the Pierce^TM^ BCA protein assay kit (#23225, Thermo Fisher Scientific), and 40 μg of each protein sample was heated at 98 °C for 5 min and then run through 8% SDS-PAGE gel. The protein samples were then transferred to a 0.22 μm PVDF membrane (#GE10600021, GE Healthcare Life Sciences, Solingen, Germany) and blocked with 5% non-fat milk in 0.1% TBS-T for 1 h. The blotted membrane was probed using specific antibodies against COX-2 (#sc-166475, Santa Cruz Biotechnology, Dallas, TX, USA) and iNOS (#13120s, Cell Signaling Technology, Danvers, MA, USA), 1:1000 dilution with 5% non-fat milk in 0.1% TBS-T at 4 °C overnight. This was followed by washing thrice with 0.1% TBS-T solution for 10 min and probing with secondary antibodies (goat anti-rabbit IgG [H + L] secondary antibody, HRP, #31460, 1:5000 dilution, Thermo Fisher Scientific) for 2 h at RT. The blots were developed using Pierce™ ECL Western blotting Substrate (#32106, Thermo Fisher Scientific), visualized using Alliance Q9 mini (Cambridge, UK), and quantified using ImageJ software 1.52a (National Institutes of Health, Bethesda, MD, USA).

### 4.7. Scratch Assay

HaCaT cells were seeded in six-well plates and incubated until they reached confluence. Scratches were developed using a sterile pipette tip, and the cells were washed twice with PBS to remove loose cells and debris. The cells were treated with *n*-hexane extract (1 μg/mL), friedelin (0.04 μM), and lupeol (0.2 μM). Wound area progression was observed using a light microscope (Nikon Ti-U; Nikon Instruments, Tokyo, Japan). The wound area was determined using the ImageJ software with the wound healing size tool plugin for ImageJ/Fiji^®^ [32].

### 4.8. Antibody Array

The RayBio ^®^ C-Series Human MMP Antibody Array C1 was purchased from RayBiotech (Peachtree Corners, GA, USA). HaCaT cells were seeded in six-well plates until they reached 80% confluency. Cells were treated with 0.04 μM friedelin and 1% DMSO in serum-free medium for 24 h. The conditioned medium was collected and centrifuged at 4500 rpm for 5 min to remove cell debris. The antibody array was prepared according to the manufacturer’s instructions. The membranes were developed for chemiluminescence detection using the Alliance Q9 advanced imaging system, and the intensity was measured using ImageJ software. Positive control spots were used to normalize the signal intensity between treatments.

### 4.9. Gel zymography Assay

HaCaT cells were seeded in six-well plates until they reached 80% confluence. The cells were treated with friedelin (0.04 and 0.4 μM) and lupeol (0.2 and 2 μM) in DMEM with 5% FBS for 24 h and 48 h. Gelatinase activity was determined according to a previous study [33]. Conditioned media were collected and spun down to remove cell debris. The protein from the conditioned medium was separated using gel electrophoresis in a 0.5% gelatin–8% acrylamide gel and then washed twice with 2.5% Triton X-100 solution (#T8787, Sigma-Aldrich) for 60 min. The gel was incubated with enzyme activating buffer (0.05 M Tris, 0.02 M NaCl, 0.005 M CaCl_2_, and 0.02% sodium azide) at 37 °C overnight. It was then stained using 0.2% Coomassie blue (Coomassie blue R 250 solution, #CBC006, LPS solution, Daejeon, Republic of Korea) and washed with de-staining buffer (10% acetic acid in 40% methanol) until the clear band was visible. The image was captured using the Alliance Q9 advanced imaging system, and the gelatinase activity was measured using ImageJ software.

### 4.10. Quantitative Reverse Transcription Polymerase Chain Reaction (qRT-PCR)

Total RNA was extracted from HaCaT cells according to the manufacturer’s protocol using TRIzol reagent (#15596018; Invitrogen, MA, USA). RNA (1 µg) was reverse transcribed using the Verso cDNA Synthesis Kit (Thermo Fisher Scientific). The cDNAs were amplified using a MiniAmp Plus Thermal Cycler (A37835; Applied Biosystems, Waltham, MA, USA). Target gene expression was examined through qRT-PCR using SYBR Green reagents (PowerUp SYBR Green Master Mix, A25741, Applied Biosystems) with a QuantStudio 1 real-time PCR system (Applied Biosystems), according to the manufacturer’s instructions. The list of primer sequences for each specific gene is shown in Table 1.

RAW 264.7 macrophage cells were seeded in a 35 mm cell culture dish until the confluency reached 80%. The cells were then treated with 1 μg/mL lipopolysaccharide (LPS; #L4391, Sigma-Aldrich) in serum-free medium for 1 h to induce inflammation. Subsequently, the cells were treated with 2.5 μg/mL and 20 μg/mL of *n*-hexane extract, friedelin, and lupeol for 18 h. The total RNA was extracted using a PureLink™ RNA Mini Kit (#12183020, Invitrogen). Two microgram of the total RNA was converted to cDNA using a Tetro cDNA Synthesis Kit (#BIO-65042, Bioline, Toronto, Canada) and a Thermal Cycler Labcycler (Senso Quest, Göttingen, Germany). To determine mRNA expression, cDNA was mixed with the SensiFAST™ SYBR No-ROX Kit (#BIO-98005, Bioline, Toronto, Canada) and amplification was performed using iCycler IQ5 (Bio-rad, Hercules, CA, USA). The mRNA expression levels of target genes were normalized to that of *Gapdh*. The relative gene expression was calculated using the comparative Ct (2^−ΔΔCt^) method. The primer sequences for each specific gene are shown in Table 2.

### 4.11. Statistical Analysis

Data were expressed as mean ± SD or SEM from at least three independent experiments. Statistical analyses were performed using GraphPad Prism 9.0.0 software. Significant differences were indicated by * *p* < 0.05, ** *p* < 0.01, *** *p* < 0.001, and **** *p* < 0.0001.

## Figures and Tables

**Figure 1 molecules-27-08540-f001:**
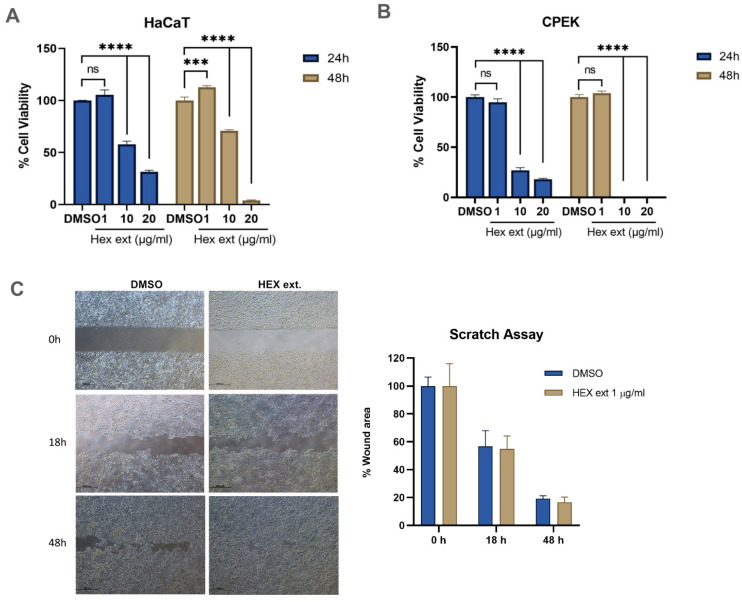
Cell growth inhibition by *H. integrifolia* in human and canine keratinocytes. (**A**) Human keratinocyte (HaCaT) cells and (**B**) canine keratinocyte (CPEK) cells treated with various concentrations of *H. integrifolia n*-hexane extract. The percentage of cell viability is presented as the mean of cell viability with standard error of the mean (SEM) compared to the DMSO treatment. Statistical significance is indicated as *** *p* < 0.001, **** *p* < 0.0001, ns = not significant. (**C**) The effect of 1 μg/mL of *n*-hexane extracts on wound healing using the scratch assay.

**Figure 2 molecules-27-08540-f002:**
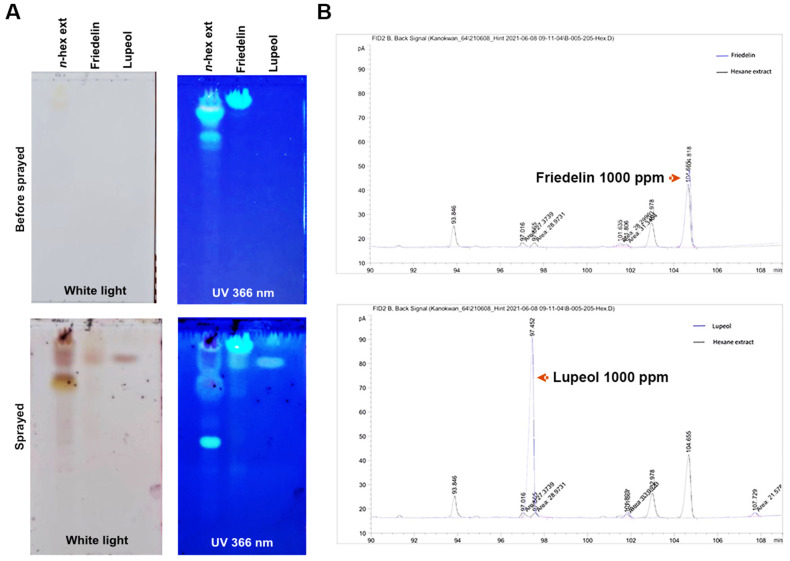
Chemical analysis of fresh stem bark *n*-hexane extract of *H. integrifolia*. (**A**) Thin-layer chromatography (TLC) shows two triterpenoids, friedelin and lupeol, in fresh stem bark *n*-hexane extraction of *H. integrifolia*. The upper panel indicates the TLC sheet before being sprayed with anisaldehyde-sulfuric acid reagent, whereas the lower panel presents the TLC sheet after being sprayed. (**B**) Gas chromatograph of friedelin and lupeol in the *n*-hexane crude extract of *H. integrifolia*. The blue line represents a peak of the standard compound.

**Figure 3 molecules-27-08540-f003:**
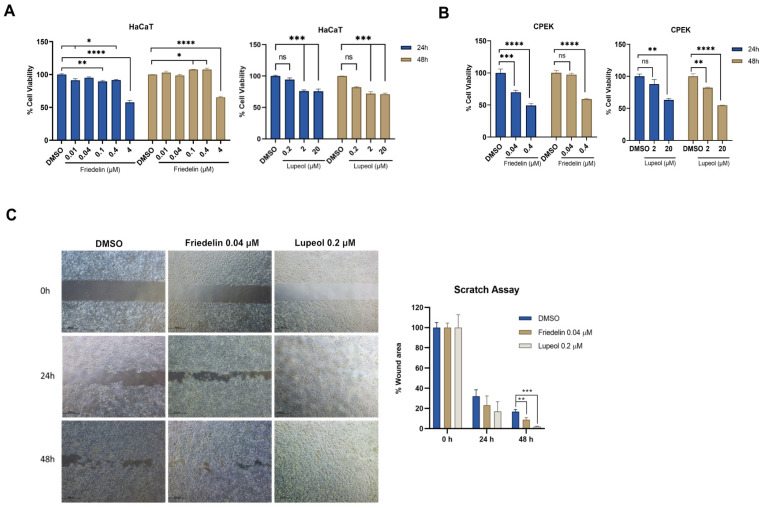
The cell viability of friedelin and lupeol on human and canine keratinocyte cells. Cell viability of human keratinocyte cells (**A**) and canine keratinocyte cells (**B**) treated with various concentrations of friedelin and lupeol. The percentage of cell viability was calculated as the percentage of DMSO treatment. The mean of cell viability is plotted with mean ± SEM. Statistical analysis was performed using one-way ANOVA, * *p* < 0.05, ** *p* < 0.01, *** *p* < 0.001, **** *p* < 0.0001. ns = not significant. (**C**) The effect of friedelin (0.04 μM) and lupeol (0.2 μM), on wound healing using scratch assay at 0, 24, and 48 h. The adjacent graph represents the percentage of the wound area. Statistical significance is indicated as ** *p* < 0.01, *** *p* < 0.001, ns = not significant compared to the control.

**Figure 4 molecules-27-08540-f004:**
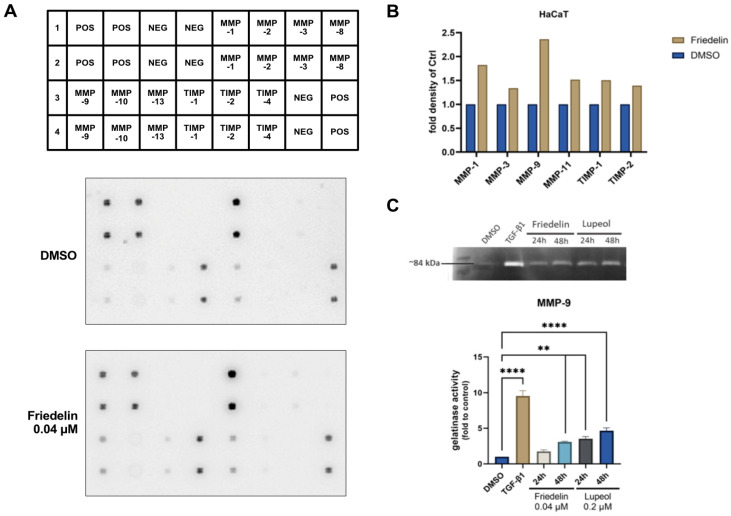
Antibody array analysis using friedelin-treated HaCaT cells. HaCaT cells were treated with 0.04 μM friedelin for 24 h; the conditioned media were harvested; and the antibody array was conducted using the human matrix-metalloproteinase (MMPs) family array. (**A**) The membrane indicates the original membrane after hybridization with conditioned media from either control-treated or friedelin-treated cell. (**B**) The graph represents the fold increase in MMPs during the MMP array, which were calculated compared to the control-treated sample. (**C**) The effect of friedelin and lupeol on MMP-9 activity was determined using gelatin zymography. The graph presents the MMP-9 activity compared to the control treatment. Statistical analysis was performed using one-way ANOVA, ** *p* < 0.01, **** *p* < 0.0001.

**Figure 5 molecules-27-08540-f005:**
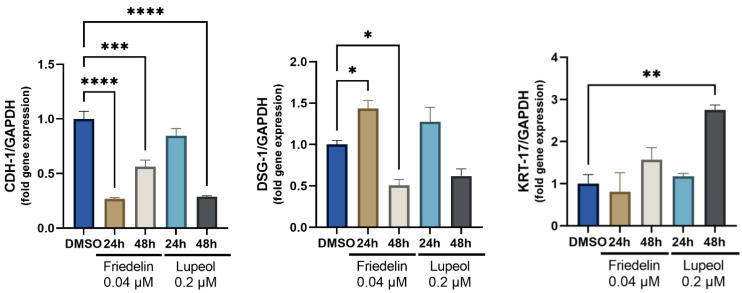
The effect of friedelin and lupeol on the expression of cell–cell adhesion genes (cadherin 1 [*CDH-1*] and desmoglein 1 [*DSG-1*]) and keratinocyte proliferative gene (keratin 17 [*KRT17*]). HaCaT cells were treated with indicated compounds for 24 and 48 h, respectively. The relative gene expression was calculated with housekeeping genes (*Gapdh)* as internalized control and the treated sample was compared with the DMSO control. Mean of fold expression to control are presented with Standard Error of the Mean (SEM). Statistical significance is indicated as * *p* < 0.05, ** *p* < 0.01, *** *p* < 0.001, **** *p* < 0.0001.

**Figure 6 molecules-27-08540-f006:**
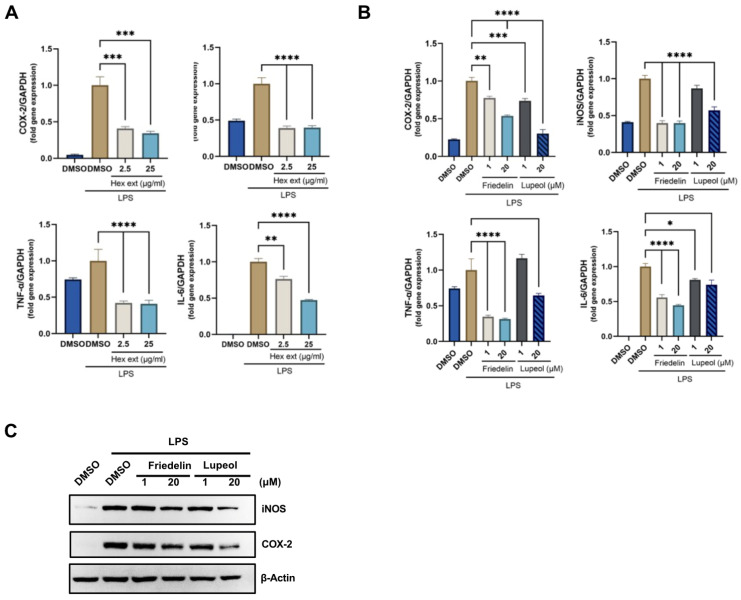
Effect of *H. integrifolia n*-hexane extract and its bioactive compounds, friedelin and lupeol, on anti-inflammatory activity using RAW 264.7 macrophage cells stimulated by LPS. The *H. integrifolia n*-hexane crude extracts (**A**) and its bioactive compounds friedelin and lupeol (**B**) were treated for 24 h. Anti-inflammatory gene expression, including COX-2, iNOS, TNF-α, and IL-6, was measured using qRT-PCR. The expression was plotted as mean ± SEM. (**C**) Western blot analysis of the anti-inflammatory effect on compound treated LPS induced RAW cells. After treatment with compounds for 24 h, immunoblotting was conducted using COX-2, iNOS antibodies. The results demonstrate the decrease in inflammatory marker proteins in dose-dependent manners. This blot is representative of three independent experiments. Statistical significance is indicated as * *p* < 0.05, ** *p* < 0.01, *** *p* < 0.001, **** *p* < 0.0001.

**Table 1 molecules-27-08540-t001:** List of the human primers used in qRT-PCR.

human KRT-17	sense strand	5′-GAGATTGCCACCTACCGCC-3′
	anti-sense strand	5′-ACCTCTTCCACAATGGTACGC-3′
human DSG-1	sense strand	5′-GAGATTGCCACCTACCGCC-3′
	anti-sense strand	5′-ACCTCTTCCACAATGGTACGC-3′
human CDH-1	sense strand	5′-GCTGGACCGAGAGAGTTTCC-3′
	anti-sense strand	5′-CAAAATCCAAGCCCGTGGTG-3′
human GAPDH	sense strand	5′-GTCTCCTCTGACTTCAACAGCG-3′
	anti-sense strand	5′-ACCACCCTGTTGCTGTAGCCAA-3′

**Table 2 molecules-27-08540-t002:** List of the mouse primers used in qRT-PCR.

mouse COX-2	sense strand	5′–CCCCCACAGTCAAAGACACT-3′
	anti-sense strand	5′–GAGTCCATGTTCCAGGAGGA-3′
mouse iNOS	sense strand	5′–GTCTTGCAAGCTGATGGTC-3′
	anti-sense strand	5′–CATGATGGTCACATTCTGC-3′
mouse TNF-α	sense strand	5′–GCCTCTTCTCATTCCTGCTTG-3′
	anti-sense strand	5′–CTGATGAGAGGGAGGCCATT-3′
mouse IL-6	sense strand	5′–TACCACTTCACAAGTCGGAGGC-3′
	anti-sense strand	5′–CTGCAAGTGCATCATCGTTGTTC-3′
mouse GAPDH	sense strand	5′–CAGGAGCGAGACCCCACTAACAT-3′
	anti-sense strand	5′–GTCAGATCCACGACGGACACATT-3′

## Data Availability

Not applicable.

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
