# Peer review of "Effect of Holoptelea integrifolia (Roxb.) Planch. n-Hexane Extract and Its Bioactive Compounds on Wound Healing and Anti-Inflammatory Activity"

_molecules, 2022, doi:10.3390/molecules27238540_

Round 1

Reviewer 1 Report

Dear authors.

I rate the work presented very highly. It stands out for the thoroughness of its execution and study design, its quality (especially the perfectly executed scratch test). The work deals with the important topic of the use of natural compounds in wound healing. Difficult healing wounds are an important medical problem that is particularly associated with diseases of civilization. The study was designed with special care, the appropriate models were selected and described in detail. The results are presented clearly. The only comment that comes to mind after reading would be to separate the conclusions as a separate chapter and expand on them. I think it would be worthwhile to carry out further studies on the antibacterial activity of the compounds in the future.

Congratulations on your results and a very well-prepared study. 

Author Response

Thank you for your review and suggestion. We changed the order of paragraphs, according to the Figure order. As per the manuscript template, the manuscript has only discussion section, not obviously separate conclusion section from discussion, therefore we had written it in the new paragraph instead.

Reviewer 2 Report

The authors of their article set new trends in science. Both test compounds are bioactive and significantly reduce wound surface area and keratinocyte migration by increasing the production of matrix metalloproteinases-9. Of particular note is the expression of cadherin 1 and desmoglobin 1 mRNA genes in keratinocyte proliferation and keratin-17 homeostasis analyzed in human immortalized keratinocyte cells. Particularly noteworthy is the research on the expression of inflammatory genes (cyclooxygenase-2 and nitric oxide synthase-induced) and pro-inflammatory cytokine genes (tumor necrosis factor-alpha and interleukin-6) by treatment with n-hexane extract from H. integrifolia and its bioactive compounds. Results showed that H. integrifolia extract and its bioactive compounds, Friedelin and Lupeol, exhibit wound-healing effects with anti-inflammatory properties mediated by regulating the expression of genes involved in skin re-epithelialization. The work thanks to this research deserves my sincere appreciation and accepts it in its present form without any significant corrections.

Author Response

Answer: We appreciate your review and comments. Hopefully, this work will be provided some benefits for those who are interested in this field.